# A Comparative Study of Urban Spatial Characteristics of the Capitals of Tang and Song Dynasties Based on Space Syntax

Liran Yin [1] , Tao Wang [1,*] and Kemi Adeyeye [2]

1   Faculty of Art and Design, Beijing Institute of Technology, Beijing 100081, China; lia.yin.39@gmail.com
2   Department of Architecture and Civil Engineering, University of Bath, Claverton Down, Bath BA2 7AY, UK; ka534@bath.ac.uk
*   Correspondence: wangtao1020@126.com; Tel.: +86-1861-831-7060

**Abstract:** Space syntax has been widely used in studies with historical components to developing a common analytical language for the comparative study of urban morphology across time and space by visual diagrams. This paper uses space syntax to analyse the inner and outer city parts of the daily life of residents in the capital cities of two dynasties, Tang and Song, to reveal the impact of changes in urban planning on the overall spatial structure of the city, the structure of commercial space, and the role of urban squares in the two dynasties under centralised rule. Based on the quantitative analysis, the results show significant differences between the Tang and Song dynasties in all three aspects of comparison. The changes in the Tang and Song dynasties' capital cities result from the interaction between the materiality of the ancient Chinese capital city form and the spatial function of the city, and the analysis of space syntax is useful for interpreting their relevance.

**Keywords:** depthmap; space syntax; spatial configuration; urban spatial structure; urban transformation

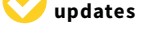



## 1. Introduction

Metropolitan studies are the specialised study of historical and contemporary cities and metropolitan regions. Within this domain is the study of the capitals of historic dynasties. Thus, the dramatic changes during the Tang and Song dynasties have attracted the attention of scholars both at home and abroad, resulting in a considerable number of research monographs. The paper presents a thematic study of the layout, form, and planning techniques of the Tang and Song dynasties. This study is derived from the archaeological and documentary data on the layout and social structure of these ancient capitals [1–3]. Previous studies on this context include that of Chang'an City of Tang [4,5], the Dongjing City of Song [6–9], as well as comparative studies of the two cities [10–12]. These studies suggest that the urban space form evolved from a closed to an open state [13] and led to a shift in the layout of urban spaces in the Tang and Song capitals. This was attributed to the collapse of the Fang system due to the closed "market" being replaced by a new market system. These studies of the Tang and Song capital cities typically employ a multidisciplinary approach and are characterised by the integration of economic [14–16], sociological and anthropological theories. They also tend to use qualitative rather than quantitative methods of analysis to explore the factors that influenced the changes in the spatial structure of the cities and the manifestations of the characteristics of the cities after the changes. However, the lack of historical documents and incomplete archaeological data has led to a lack of basis for these deductions, making it impossible to make a rational connection between the cause and effect on the overall urban spatial structure.

In recent decades, the increase in computing power has led to the development of a number of analytical tools for urban space analysis [17–19]. These include space syntax [20–22], spatial matrices [23,24] and regression models [25–27], each with its focus. Spatial matrices used for the analysis of building mass density and land use and the

regression model for predictive analysis are not applicable in the context of this paper. Space syntax is a method for describing the relationship between physical space and social elements. It is of great value in predicting human behaviour in space in urban and built environments. Diagrams can be used to visually present data that represent the associations between spatial and social elements. It also enables the underlying spatial mechanisms and socio-economic phenomena to be analysed, described and explained using exploratory spatial models [28,29]. This is precisely the link between cause and effect that was lacking in previous studies of ancient capitals. Space syntax starts from studying the topological relations of space rather than measuring it. This overcomes the inability to analyse ancient cities due to the lack of archaeological information. Griffiths explored the prospect of applying spatial syntax to the study of historical cities [30]. Over the last 30 years, there have been several attempts to study ancient spaces using spatial syntax. By combining archaeology with present-day urban planning theory and using space syntax to analyse ancient spaces [31,32]. Some Chinese scholars have utilised space syntax to interpret the relationships between ancient garden spaces [33–35]. Zhu Jianfei used spatial syntax to study Beijing City during the Ming and Qing dynasties [36]. However, no current research results have used space syntax to study the capital cities of China in the Tang and Song dynasties, so this area is still a gap.

This paper, which utilises existing research evidence on the capital city structures of Chang'an in Tang Dynasty and Dongjing in Northern Song Dynasty, (i) establishes space syntax situational perception diagrams based on the urban street network, (ii) interprets the three dimensions of the overall urban, commercial and economic morphology, and the urban square, and (iii) analyses the correlation between the social elements and physical space evolution of the two ancient capital cities. Finally, the commercial and population distribution of the two capital cities are compared with depthmap calculations used to verify the correctness of the analysis process.

## 2. Overview of the Research Object

The capital city is the epitome of social changes, and the changes in various levels of social ideology are directly reflected in the urban spatial structure of the capital. This paper takes Chang'an of the Tang Dynasty (henceforth: Chang'an) and Dongjing of the Northern Song Dynasty (henceforth: Dongjing) as the research objects to understand the rationale behind the formation and evolution of the two cities.

### 2.1. Overview of Chang'an of the Tang Dynasty

Chang'an was the Tang Dynasty's capital, which gradually grew and expanded, based on the Daxing city of the Sui Dynasty. Chang'an had three layers, the outer city, the imperial city and the imperial palace. The overall layout of the capital was a neat square on plan (Figure 1). With the Imperial Palace as the centre, the capital city had a central axisymmetric layout. The central axis passed through the Mingde Gate in the outer southern city, the Zhuque Gate in the Imperial City, the Chengtianmen Gate in the Imperial Palace and the Taiji Hall. It ran through 11 streets and 14 east-west streets. The Fang, Square or Block, as an urban grass-roots community unit within Chang'an City, was the product of a strict urban management system. With streets subdivided into residential spaces, temples, government offices, and mansions. The market was the centre of Chang'an's economic development during the Tang Dynasty, and there were two markets, east and west, which were symmetrically distributed in the outer city [37].

### 2.2. Overview of Dongjing of the Northern Song Dynasty

Dongjing was the capital of the Northern Song Dynasty, and its predecessor was the Bianzhou City of the Tang Dynasty. Moreover, Bianzhou was rebuilt as the capital in the later Jin Dynasty, the Later Liang Dynasty, the later Han Dynasty and the Later Zhou Dynasty. In the Northern Song Dynasty, it was built and expanded many times, forming a prosperous Dongjing City in its mature period. Dongjing City consisted of three layers,

the outer city, the inner city and the Imperial City. The plan layout of Dongjing City was rectangular, but it was not as neat as that of Chang'an City. It was more like a diamond shape (Figure 2), and the direction of the city walls hadve a slight twist. Dongjing City also had an axis running from north to south. The whole city was generally in axial symmetry; from the Nanxun Gate in the outer city to the Zhuque Gate of the inner city and the Xuande Gate of the Imperial City. The open market system in Dongjing City broke through the impenetrable walls, resulting in a more flexible distribution of streets and lanes and more prosperous commercial development in the City [38].

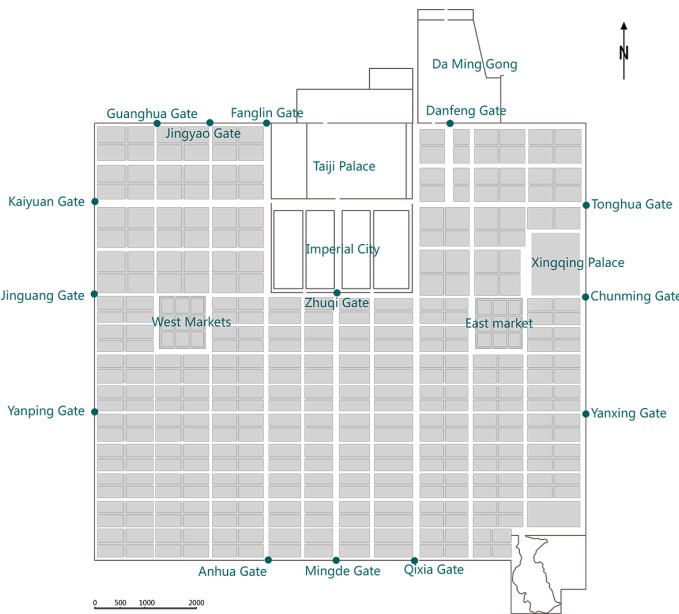

**Figure 1.** Recovery plan of Chang'an of the Tang Dynasty. Drawing by the author based on Liu Dunzhen: The History of Ancient Chinese Architecture. China Architecture and Building Press, 2008, p. 118.

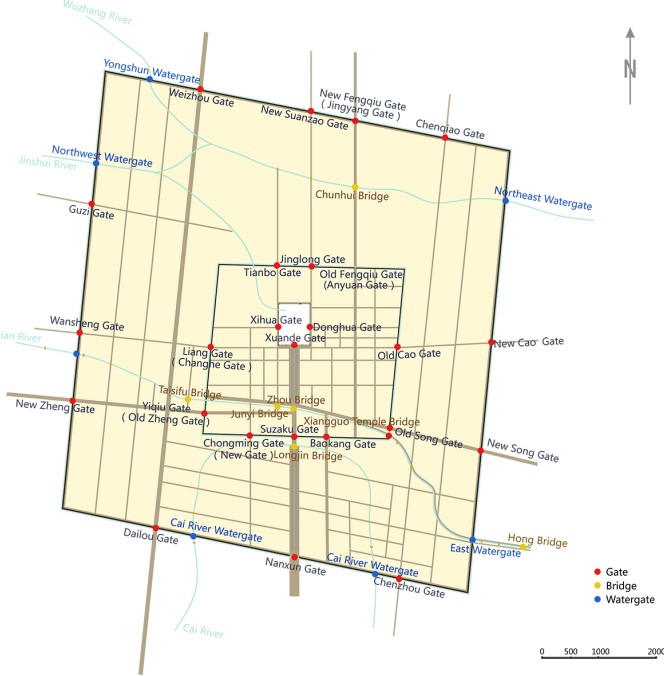

**Figure 2.** Recovery plan of Dongjing city of the Song Dynasty based on Dongjing Menghua Lu and the papers by Ye et al.

### 3. Overview of the Theory of Space Syntax

Space syntax is a set of theories and tools for spatial morphological analysis proposed by Hillier and Hanson in the 1970s [39]. It emphasises the ontology and importance of space, using the mathematical theories and methods of graph theory, relying on space perception by people in motion in real environments, for the quantitative description of urban and architectural spatial structure morphology [40]. The objects of study include cities, settlements, buildings and other human spaces. Space syntax reflects space with social meaning and can provide a direct answer to the social logic of space [41]. The core concept of space syntax is spatial configuration, which means the spatial elements' interrelationship in a spatial system [42]. Space syntax generates the embodied diagrams through depthmap, which is the fusion of comprehensive spatial information and simplifying and interpreting complex urban information. This constitutes the relationship between spatial cognition and spatial configuration [43,44]. Depthmap is mainly a computer program for spatial network analysis of buildings and urban systems created by Alasdair Turner, and is one of the computer analysis tools for space syntax [45–47].

Based on the principle of spatial cognition, there are three basic spatial segmentation methods for space syntax: convex space, axial line and visibility graph analysis. In this paper, the axial line and the visibility graph analysis method are used.

The axial analysis simplifies space into lines, which represent linear space and correspond to the linear movement of people [48,49]. Travelling in the direction of the axis is the most economical and convenient way to move, and it is also the way people move instinctively, so the axis has both the dual meaning of a line of sight and a motion state [50].

An isovist is the area in a spatial environment directly visible from a location within the space. The isovist concept has a long history in architecture, geography, and mathematics [51–53], and Benedikt introduced a set of analytic measurements of isovist properties that promoted the use of the isovist as a way to deal with the relationship between the viewer and the spatial environment [54]. Turner et al. developed the isovist analysis using a regular grid form, which formed the Visibility Graph Analysis now used in space syntax [55,56].

The essential syntactic variables used in this paper include connectivity value, integration value, intelligibility and synergy.

The connectivity is expressed in terms of the number of nodes adjacent to a node as the node's connection value. In practical spatial systems, the higher space's connectivity value, the better the spatial permeability [57].

Integration describes the average depth from any space of origin to all other spaces in the system.

Intelligibility is a linear correlation value indicating the degree of connectivity and global integration. "Intelligibility means what we can see from the spaces that make up a system, i.e., how many other spaces are connected. Furthermore, to what extent it is an excellent guide to what we cannot see, i.e., how well each space integrates with the whole spatial system. An intelligible system is a space that is well connected and often well integrated. An incomprehensible system is one in which well-connected spaces are not so well integrated that what we can see of their connections misleads us about the place of that space in the system as a whole [58]".

Synergy is the linear correlation value between the global integration value and the local integration value [59].

### 4. Methods

This chapter aims to use space syntax to understand how the city's spatial street network and commercial spatial layout have evolved under the different urban plans and what contrasts they have produced. Different urban spatial patterns emerge in ancient cities due to political and economic factors. However, the analysis of their overall spatial patterns from the perspective of syntax revealed isomorphism in the sense of topology, making it possible to compare different cities. This chapter is divided into three sections.

Each is based on a different perspective to compare and analyse the Tang and Song capitals, using depthmap to carry out an axial analysis in the first section. A visibility graph analysis in the second and third sections, so that the different cities form different chromatographic layers, can clearly distinguish the differences in the two cities' overall spatial structure.

When constructing the axial graph and visibility graph of the city in CAD, should exclude the palace and imperial city parts from the Chang'an and the Dongjing City. The palace and imperial city existed as political functional spaces. In contrast, this study took the study of the city's daily urban area as the main aspect, and the political area was not the main object of study. Moreover, one of the functions of the imperial city itself was to serve as a residential space for the royal family, which was a relatively independent space, and was a relatively closed space in daily life, unless under special circumstances, such as when the royal family was summoned to the court. Therefore, when carrying out the space syntax analysis, the palace city and the imperial city were considered a whole architectural body space to not interfere with other streets.

*4.1. Comparative Analysis of the Structural Features*

In this section, the axial line is used to calculate the degree of global integration (R = n), local integration (R = 3), synergy and intelligibility. The results of analyses are then used to interpret the overall urban spatial structure. The axial line from red to blue in the axial view represent the strongest to weakest global integration. The axes in the top 10% of global integration are called integration cores, and the form of the compact core reflects the basic form of a city. The urban compact core is the centre where people flow together in the whole city. Due to its dominant position in terms of accessibility, the compact core has far stronger social and economic functions than other regions, thus becoming the centre of social activities in the city.

The constructing of the Chang'an City axial graph required that each Fang (Square) be drawn as a single building. As the basic unit of Chang'an City's urban organisation and management, the Fang was relatively closed, with only two or four exits, and there are many lanes inside the Fang. Still, only a few were connected to the street outside the Fang. This way of drawing allowed a more intuitive view of the global integration of the entire outer city.

The global integration of Chang'an City in Figure 3 shows that the highest integration value of Chang'an City was the two east-west streets adjacent to the East-West Markets. Chunmingmen Avenue, between Chunmingmen and Jinguangmen, was an essential lifeline for the East-West Avenue through the city. The Imperial City and Xingqing Palace was representing political significance on the north side. The East-West Markets were holding the economic lifeline on the south side, and Chunmingmen Gate was leading eastward through the Hangu Pass to the East Capital, through which grain and other commodities from the economically developed areas in the Yellow River Valley and Yangtze River Valley were transferred west, from Jinguangmen up to Sichuan, Yunnan, Gansu, Long and the Western countries. Envoys, merchants and commodities from countries west of the Guanlong Region and Western Asia also came to Chang'an via this road [60]. Therefore, Chunmingmen Street is of traffic significance both from the political and economic perspectives. The second-highest level of integration across the city was on the south side of the East-West Markets. This high level of integration was due to being on the south side of the street immediately adjacent to the eastern and western markets, which also served as a diversion for Chunmingmen Street.

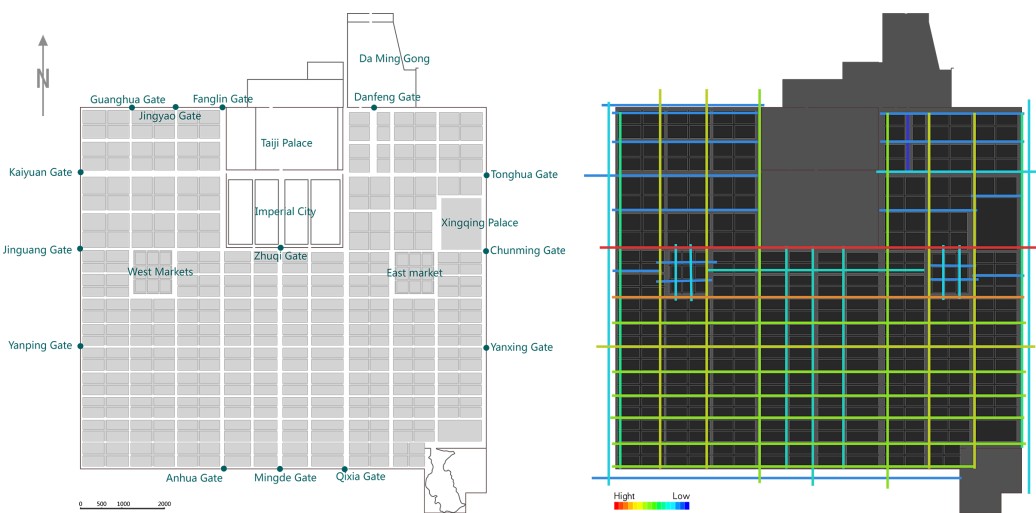

**Figure 3.** Global integration of the axes in Chang'an of the Tang Dynasty.

According to the global integration value of the Dongjing City in Figure 4, there were four significant red axes, including the New Fengqumen Street to Mahang Street, the street from Liang Gate to Old Cao Gate, Bianhe Avenue and the South-north Street from Xingguosi Bridge to the Chongming Gate. These were the main roads with good accessibility to link the central and peripheral areas. These streets were densely populated with commercial shops, close to the political centre, which constitutes the core space of Dongjing. The degree of integration in the periphery of the capital city was low, which may be because the outer city was planned with more military barracks and granaries located close to the outer city gates or water gates. This resulted in a less dense distribution of the road network, with poor accessibility and reduced population concentration compared to the commercial areas of the inner city.

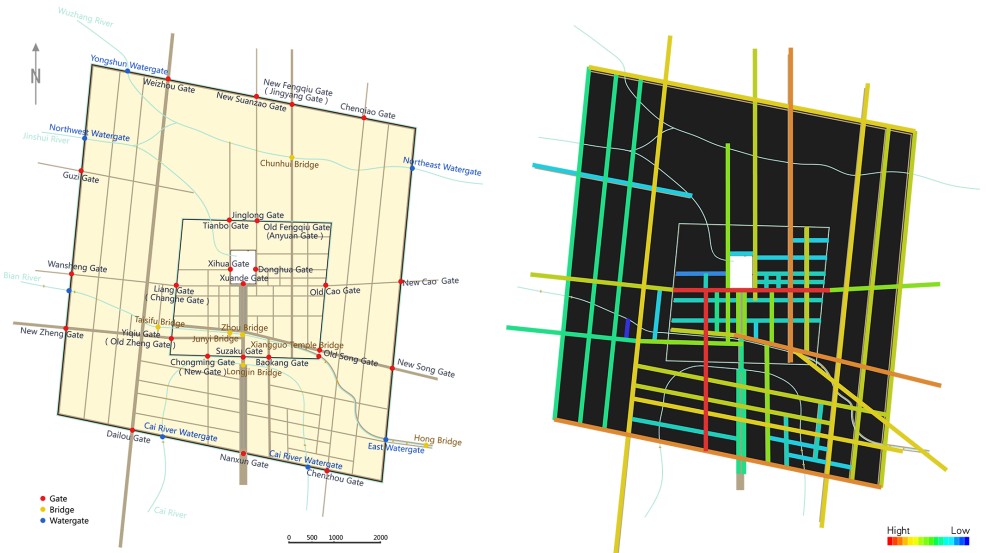

**Figure 4.** Global integration value of the axes in Dongjing of the Song Dynasty.

The global integration degree calculation was affected by the incomplete archaeological and documentary data on both cities. Therefore, the local integration degree was applied for supplementary analysis. Local integration (radius = 3) reflects pedestrian movement. Before the analysis, it was necessary to verify the urban space's local and global coordination through the synergy ratio. Synergy is the correlation value between the global integration value (radius = n) and the local integration value (radius = 3). It describes whether the local and global accessibility of the axes are consistent. Axial systems with

high synergy are more likely to reflect the whole through to the local. The closer $R^2$ of synergy is to 1, the more the tendency towards a single core. The closer it is to 0, the more the tendency of multiple cores, with low integrity. The ratios for the Chang'an City and Dongjing cities were 0.799 and 0.934 respectively. The results indicated that both capitals had a high synergy, that the local and global were coordinated, and that capitals' cores tended to be unified single-core spaces (Figure 5a,b).

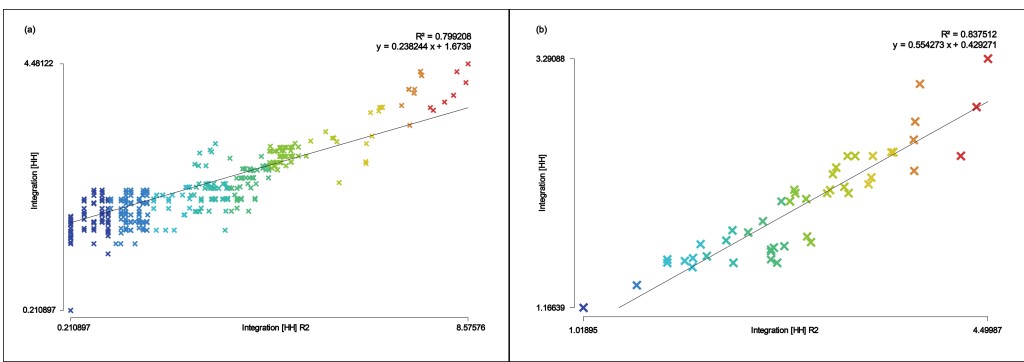

**Figure 5.** (**a**) Synergy of Chang'an of the Tang Dynasty and (**b**) synergy of Dongjing of the Northern Song Dynasty.

However, when calculating the local integration value of Chang'an, the street axes mapping previously carried out for global integration did not apply to the calculation of the local integration. This made it important to consider the street conditions and residential layout form within the Fang. There were 109 blocks as the basic urban community unit of Chang'an. There were two kinds of primary streets in the Fang, straight-lined streets and cruciform (cross) streets, with two or four exits, and the more subdivided alleys were called Qu. The traditional courtyard form was inherited from the Chang'an city, where the porch-yard style was still prevalent and the courtyard style had travelled, as seen in the Dunhuang murals and the typical courtyard in Beijing, as well as in the aerial view of the ancient city of Pingyao, which shows the arrangement of the houses (Figure 6). The house doors of the dwellings were distributed along the streets of the Fang and were neatly arranged. Therefore, the assumptions of the skeletal pattern of the streets within the block in Hui Sun's article [61] (Figure 7) were used in the space syntax analysis for Chang'an.

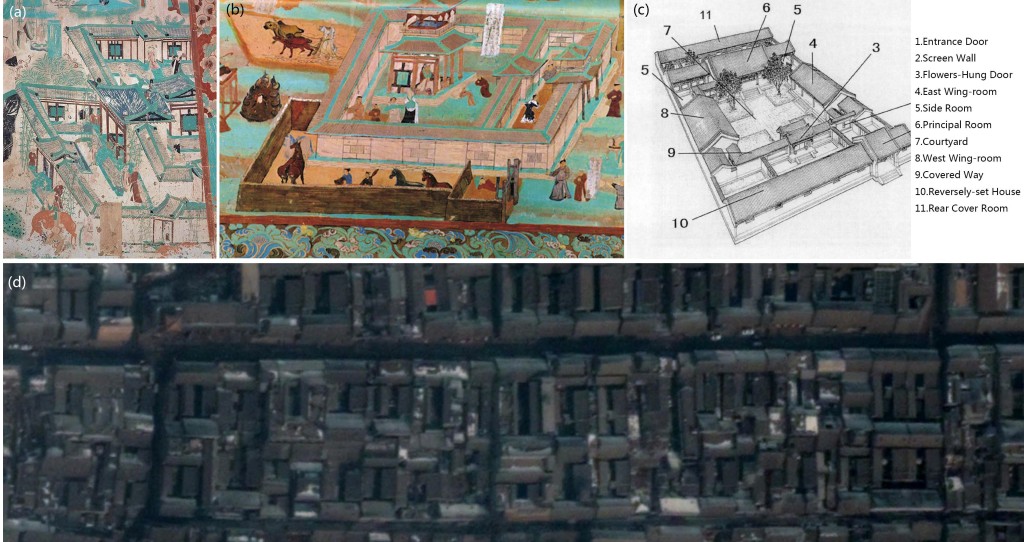

**Figure 6.** (**a**) Frescoes from Kiln 9 at Dunhuang; (**b**) frescoes from Kiln 85 at Dunhuang; (**c**) A typical three-story courtyard in Beijing [62]; (**d**) An overhead view of the ancient city of Pingyao.

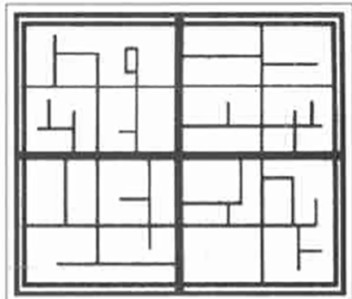

**Figure 7.** Assumed diagram of the internal street skeleton of the lanes in Chang'an of the Tang Dynasty from Hui and Jiang: Analysis on the Internal Structure of Fangli Units of Chang'an in Tang Dynasty, City Planning Review, 2003 (10), pp. 66–71.

As shown in Figure 8, the streets with higher integration value appeared on the Fang's straight-lined and cruciform cross streets. The reason may be that, according to the policy, Fangs did not have direct access or open directly to the street. The streets outside the Fang only had the function of passing through, which reduced the possibility of people staying and gathering. Thus, the relative concentration of people where the local integration is highest was in the intersections or alleyways inside the Fang, which were connected to the street bordering the Fang and the subdivision of the Qu inside the Fan.

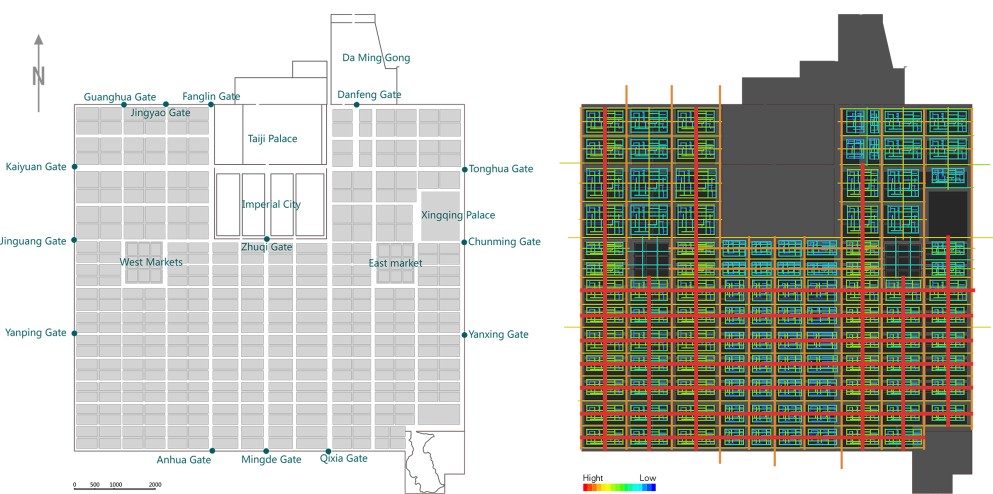

**Figure 8.** The local integration value of Chang'an of the Tang Dynasty.

The local integration value of Dongjing City almost coincided with the street with the highest global integration value (Figure 9). The street from Liang Gate to Old Cao Gate and the street from Xingguo Temple to Chongming Gate had the highest local integration values. The street from Liang Gate to Old Cao Gate, passing through the Xuande Men Square of the Imperial City, represented the royal family's power and dignity. There was the Imperial Corridor on the south side. To the east, it went to Panlou Street via Zuoye Gate, where there was Panlou, a well-known restaurant. To the south, there were the Eagle Store, pearls, silk cape and incense shops, Jieshen Alley, a financial and monetary trading centre, and Sangjiawazi, the largest entertainment centre in the south. Further east were various kinds of restaurants, tea shops and hotels on Old Caomen Street. To the west was Yonglu Street via Youye Gate, where there were many inns, pharmacies, and hotels. The street from Xingguo Temple to Chongming Gate had Xingguo Temple, one of the capital's religious centres. Many references about the Northern Song dynasty's emperors were praying for rain or burning incense at Xingguo Temple, with many official offices and palaces nearby, hence the prevalence of people gathering at these intersections.

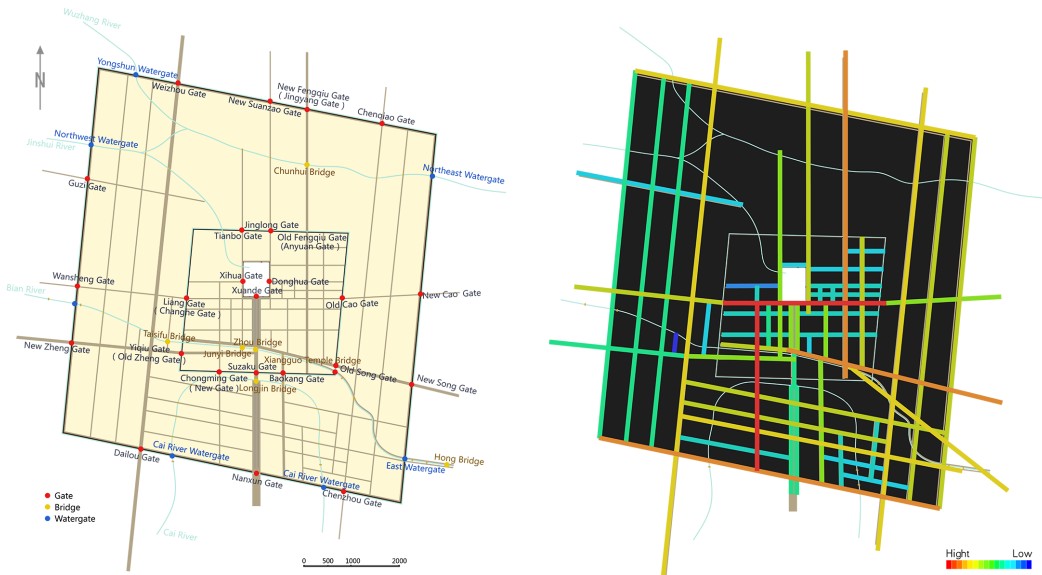

**Figure 9.** The local integration value of Dongjing of the Northern Song Dynasty.

Intelligibility is measures of axial map characterisation of global features, which together reflect the city's recognisability and can be used in a comparison between Chang'an city and Dongjing city. High intelligibility indicates that the spatial system's local structure helps to build up a picture of the whole spatial system. However, many connected spaces cannot be well integrated into the overall system for cities that lack intelligibility. Therefore, relying on these visible connections will mislead the perception of this space's position in the overall system [63]. In space syntax, intelligibility is represented by the regression coefficient $R^2$. $R^2$ values between 0 and 0.4 indicate that the coefficients are poorly intelligible. Between 0.4 and 0.7 indicate that the coefficients are intelligible, and between 0.7 and 1 indicate that the coefficients are extremely intelligible.

The intelligibility index of Chang'an is 0.170 (Figure 10a). The city with low intelligibility indicates that the local space is not closely integrated with the whole structure. The accuracy of predicting the whole through local space is low, and the local space in the Fang of Chang'an City was disconnected from the overall urban system. This result is consistent with the calculation of Chang'an City's local integration, where the lanes inside the Fang were blue, and the streets outside the Fang were red. The contrast in local integration reflected the disconnect between the lanes in the city and the city streets. The reason for this is twofold, in terms of the structure of the Fang, which was the main residential unit in Chang'an City, each Fang was rectangular with two or four gates through which the residents of the Fang must enter. The Fang was built with walls, and residents were not allowed to open the doors directly to the street, except for the mansions of senior officials, and the residents of the Fang of special circumstances that were allowed to face the street (Figure 11). A head was designated for each Fang in charge of the Fang's affairs. This regulation created a relatively closed space in terms of urban management, reducing the number of entrances, and controlling the residents' access, making it easier to manage [64]. Therefore, even though there was a rich street texture inside the Fang, there were only two or four roads that were identical to those outside the Fang, making it inconvenient for residents to enter and exit. From the perspective of urban spatial patterns, the spatial pattern of this capital was neater and tidier. It ensured the uniqueness of the Fang and market Gate entrance, but it also resulted in the Fang's local centre not being well integrated into the overall spatial structure.

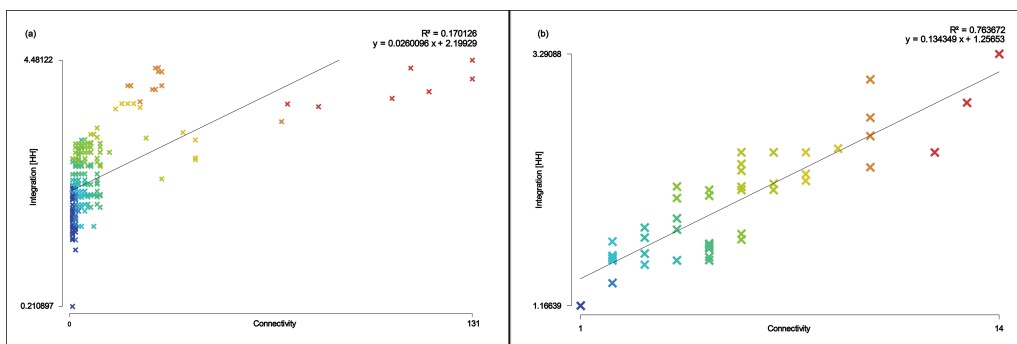

**Figure 10.** (**a**) Intelligibility of Chang'an of the Tang Dynasty and (**b**) intelligibility of Dongjing of the Northern Song Dynasty.

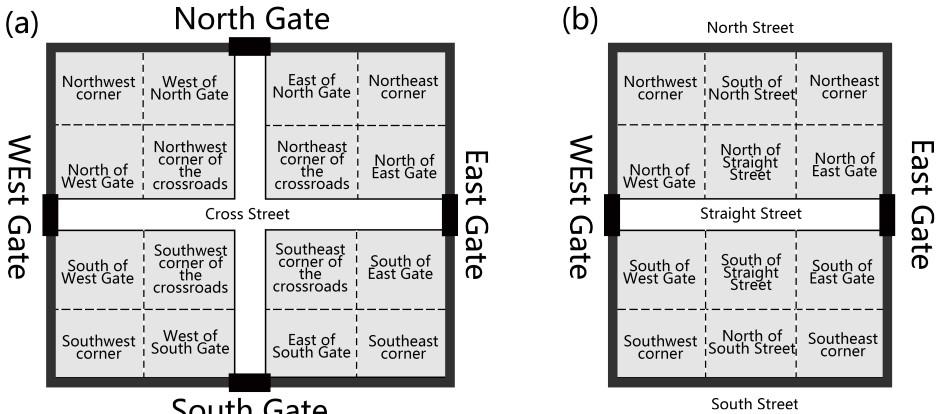

**Figure 11.** Two types of internal location map of the Fang in Chang'an [65]. (**a**) Nearly square-shaped Fangs, with doors on all sides. (**b**) Rectangular Fangs with only two side doors.

The intelligibility index of Dongjing City is 0.763 (Figure 10b). From economics and sociology, the higher the intelligibility, the better its local centrality can be integrated into the global spatial structure. Thus, it generates the multiplier effect of economic and social activities, making the spatial system more diverse and complex. The capital has a street market system formed after the collapse of the block market system. Although Dongjing City is the Northern Song Dynasty's capital and has a strong political function, its urban pattern had a certain spontaneity. Based on inheriting the original road pattern, the street has been modified, and its development follows residents' needs in the urban space. Streets and alleys are open and flexible, allowing for the rapid growth of commerce in the city and forming various markets scattered throughout the city.

*4.2. Analysis of Business Pattern*

In previous studies, the commercial space was more distributed within the highly accessible and visible regions. The accessibility along the axes were previously analysed. So was the visual integration, to understand the internal mechanism of the spatial features. Most businesses were distributed in the region with high visual integration because high visibility was an essential factor for attracting people flow. Dividing the capital into a 1*1 grid (m), and the outdoor space of the capital was used for visibility graph analysis, and Figure 12 shows the street outside the Fang near the Fang Gate. The visual integration represents the space's visibility, indicating the exterior space scope where human beings are free from the structures. The visual integration diagram went from red to blue, representing the visual integration from high to low.

In Chang'an's business layout analysis, the city was divided into the early, mid and later periods. The urban administration system was rigorous to varying degrees in

different periods, resulting in different business conditions and different regions for space syntax analysis.

In the early period of Chang'an, residences and markets were extremely close socially and materially. The eastern and western markets areas, after several surveys, were 0.924 and 0.956 square kilometres, respectively. The city's total area was 84 square kilometres; the two markets only accounted for 2.2% of the city, which was negligible. Moreover, the commercial functions of the capital were concentrated in East-West Markets. The stores other than those in the eastern and western markets are seldom recorded in the literature. For this reason, only the internal situation of the eastern and western markets are presented when studying the business space pattern in the early period of Chang'an. The two markets were divided into nine parts by well-shaped roads, and roads along the lane wall. In each part, the road was divided into several alleys. Moreover, only eight gates were retained in the entire market.

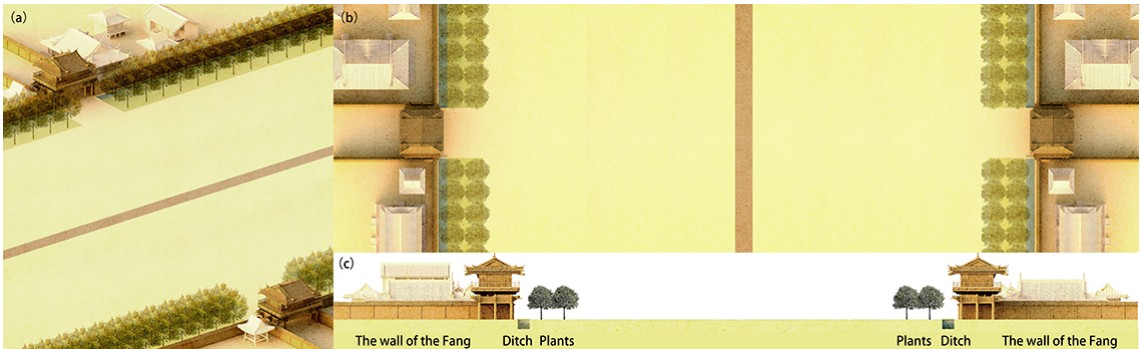

**Figure 12.** (**a**) Diagram of the streets outside the Fang near the Fang Gate in Chang'an, (**b**) top view, (**c**) section view.

The visual integration within the eastern and western markets were calculated (Figure 13). No literature and archaeological materials were available for the layout of stores and streets in the eastern and western markets, but this could be inferred from the visual integration that the intersection of the roads attracted the most flow of people. Thus, the region would have had a distribution of various stores and street peddlers.

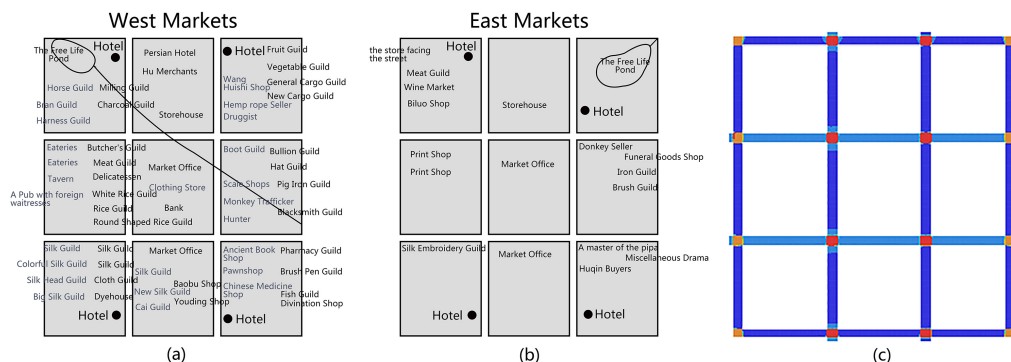

**Figure 13.** (**a**,**b**) Internal structure of the East-West Markets in Chang'an of the Tang Dynasty from Shi Nianhai: Historical Atlas of Xi'an, Xi'an Map Publishing House, 1996 and (**c**) visual integration of the east market in Chang'an of the Tang Dynasty.

In the mid and later period of Chang'an, the long-term commercial spaces were impacted by the constantly developing urban commercial economy, and the integrated system of residence and markets were also relaxed, resulting in some stores going beyond the East-West Markets' scope to appear outside the markets. The stores within the Fang were mainly engaged in catering, entertainment and accommodation. This is why a visual integration analysis of the whole city was carried out in the study of the late Tang Chang'an period.

However, as can be seen in Figure 14b, the results of the analysis did not give the desired results but reflected the urban spatial state of the city as depicted by the actual space of the Chang'an City, with high visual integration only at crossroads, and low visual integration in the regular streets. It is recorded in many places in the literature that rows of trees were planted on both sides of the roads as street plantings [66]. However, due to the wide streets of Chang'an City, the trees did not cause a significant impact on the results of the Visibility Graph Analysis. Only the visual integration in the vicinity of the trees was relatively low, and the visual integration near the trees is shown in Figure 14c. At this point, we found that space syntax was limited in some respects. In the face of such a rigid urban layout, it was impossible to conclude the changes in a single form of square streets by using space syntax tools. It was not like the modern city that we often analyse where the urban fabric is cellular. After years of subjective human behaviour, the interaction between urban structure and human movement will show up in the urban fabric. However, in the urban street network of Chang'an, which was close to absolute geometry, it was completely separated from people's life.

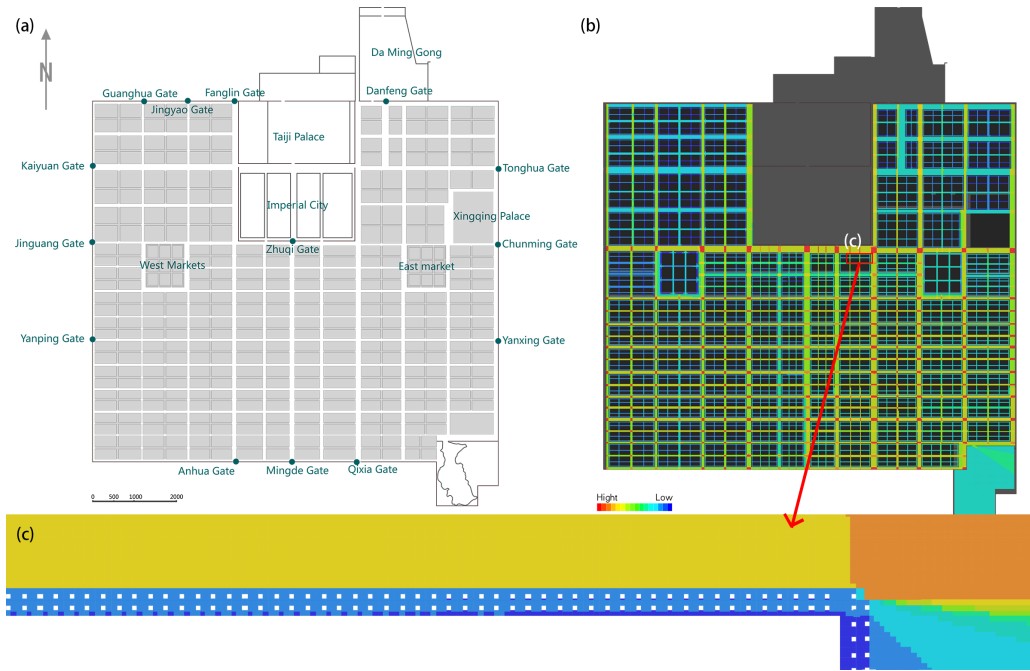

**Figure 14.** (**a**) Chang'an City plan, (**b**) City-wide visibility graph analysis, (**c**) Enlarged view of the results of the city-wide Visibility Graph Analysis around the trees on both sides of street.

The visibility graph analysis for Dongjing city was made for the whole city. Figure 15 shows Dongjing's visual integration, in which the largest region with the maximum visual integration was the street in front of the Xuande Gate (the street from Liang Gate to Old Cao Gate). The street in front of the Xuande Gate had the highest visual integration value because, compared to other spaces, it was a straight street with a wider vision. The second-largest region was the Imperial Street from Xuande Gate to the Zhou Bridge, which was the central axis of Dongjing City. It was a bustling street, with the Eastern and Western Jingyun Palaces and the central government office. On the northeast of the ZhouBridge was Daxiangguo Temple, the largest temple in Dongjing City. Next was Bianhe Avenue, where there was the eastern Imperial Street. As the most important canal transport channel in Dongjing City, Bian River had ships docked there, and it also hosted the residents of the merchants from the south, hence the presence of several mansions in this area. The famous space nodes of this street were the Daxiangguo Temple and the Shisanjian Building. Finally, high visual integration values at the crossroads could be observed from all four directions from the spatial surface. At a deeper level, the intersection space was the convergence of people flow, logistics and information flow. It was the best location to build landscapes and

engage in business activities within the city. By the cohesion of its superior commercial geographical location, the node attracted all kinds of commercial shops in this region and formed the commercial space with the nodes as the core. The streets and alleys with low visual integration were mostly located in the internal second-level streets and alleys, with fewer intersection points and more spatial turning points. In such areas, it was difficult to observe other spaces, resulting in poor visibility. Figure 16 shows a partial view of the Qingming Shanghe River, showing the commercial streets of the city of Dongjing.

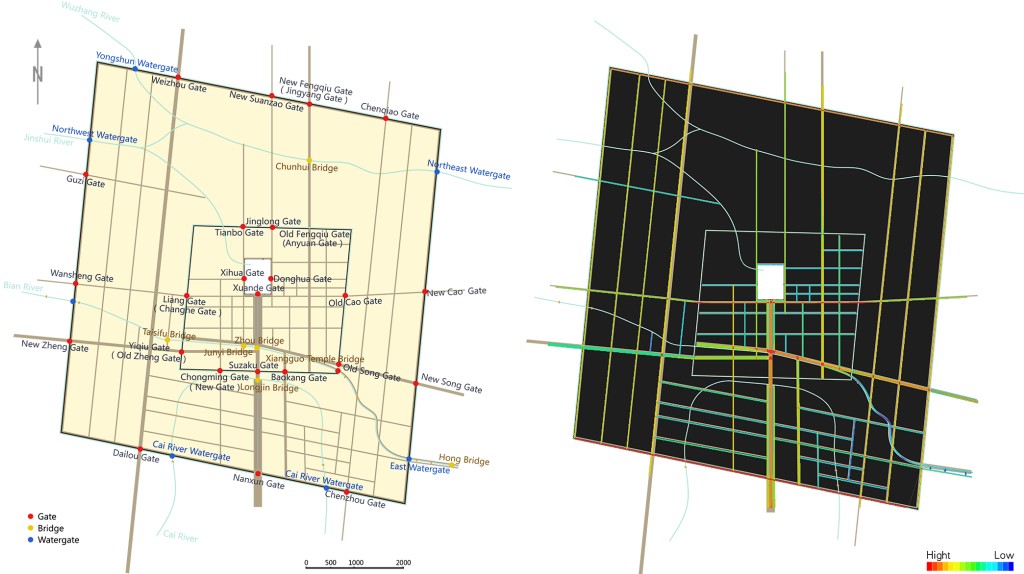

**Figure 15.** The visual integration value of Dongjing.

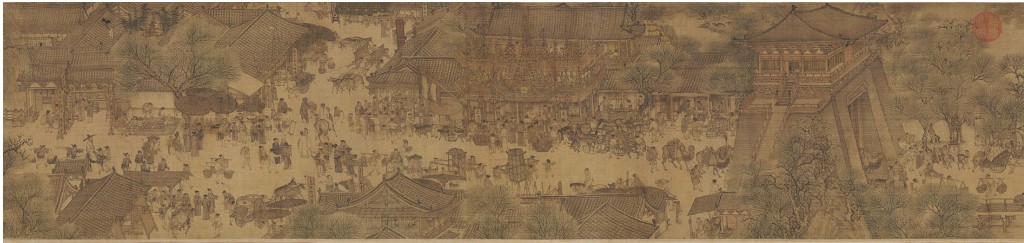

**Figure 16.** Part of the Qingming Shanghe Tu.

### 4.3. The Square

Squares are the oldest form of urban external space. Jackson argues that "squares are urban form which acts to draw people together [67]", and American scholar Kelvin Lynch argues that "squares in high-density urban centres should be the place where people come together. A square is a vibrant focal point, typically characterised by a paved ground, a strong sense of enclosure formed by buildings, or defined by, or in relation to, the street. It is characterised . . . by its ability to attract large numbers of people and to provide a place for meetings [68]". The form in which the square exists is of great importance to the city and therefore forms part of our understanding of ancient capitals.

The visible and the accessibility layers were calculated separately for the square and then analysed in the specific analysis. The visible layer refers to the area seen by the residents based on the average 1.6 m high line of sight of residents; the accessibility layer refers to all accessibility areas in the city, including the through roads, under the shade of the forest, and the city gates (excluding the nighttime situation when the inner city gates were closed). The visual integration of the accessibility layer is usually used to analyse how easy it is for the spaces to be reached in the system as a whole and how easy it is to reach other spaces in the system. The connectivity of the visibility layer (Con-eye) is used

to analyse the area in which each space can be seen by other spaces within its field of view, namely space's visibility.

The square in front of The Zhuqi Gate of Chang'an was linear and 220 meters wide. It was the widest street in the country at the time. Functionally, it served as a palace square; a place to display royal power and dignity. As Kevin Lynch said in The Good City Form, "The control of space can be obtained by force, and the control of space can also be used to demonstrate and enhance power". The court held celebrations or ceremonies in this square on the first day of the first lunar month, on the winter solstice, or when setting up and receiving foreign envoys. Now it is generally closed, and members of the public are permitted only during national festivals. Danfengmen Street in front of Daming Palace was the palace square for the Eastern Palace of the prince where the New Year's Day celebration and other ceremonies were held.

The accessibility layer's visual integration in Figure 17b indicated that the square's accessibility in front of the Zhuque Gate was better than that of other streets. In Figure 17c, the isovist connectivity value was higher than that of other streets. The intersection with other streets reached the highest, which indicated that the square had a better field of view and less occlusion than other streets.

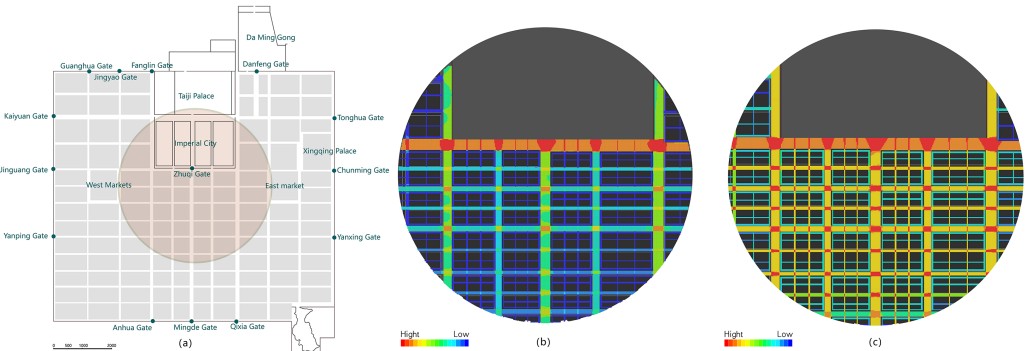

**Figure 17.** (**a**) Recovery plan of Chang'an of the Tang Dynasty, (**b**) the visual integration value of Zhuque Avenue and (**c**) the isovist connectivity value of Zhuque Avenue.

The T-shaped square in front of Xuande Gate in Dongjing was a space for royal activities and a public space where residents could be involved. During the Lantern Festival, the Hungry Ghost Festival and the Spirit Festival, the residents viewed the lanterns mainly at the square. The square was dominated by the street in front of the Xuande Gate and the Imperial Street from Xuande Gate to Zhou Bridge. The Dongjing Menghua Lu gave a detailed description of the layout of the Imperial Street in front of the Xuande Gate (Figure 18): "The Imperial Street, from Xuande Gate to the south, was about two hundred paces wide, with an imperial gallery on either side, between which the merchants used to buy and sell in the past, and it was forbidden in the Zhenghe period. Each set of black lacquer forks, and the centre of the road set two rows of red lacquer forks, and the central royal road were forbidden to walk by horses and common people. Pedestrians were outside the red lacquer fork under the corridor. There were two brick-and-stone ditches in the branches. Lotuses were planted in the Xuanhe period, peaches, pears and apricots were planted near the shore. Flowers bloomed in spring and summer [69]". So the Imperial Street was divided into three parts. This layout further emphasised the effect of the city's central axes. Based on this spatial state, the T-shaped square was calculated in two parts, the visibility layer and the accessibility layer, and then analysed.

As can be seen from the results of the connectivity of the visibility layer for the Dongjing city (Figure 19b), the results of the visual integration of Dongjing show that Imperial Street was divided into three parts. The middle Imperial Road had a higher integration, and the central position had a better view and was easier to see. Therefore, they were the locations with the highest integration in the entire road network of the square,

from the crossroads in front of the Xuande Gate to the Zhou Bridge, and even across the Zhou Bridge all the way south to the Longjin Bridge.

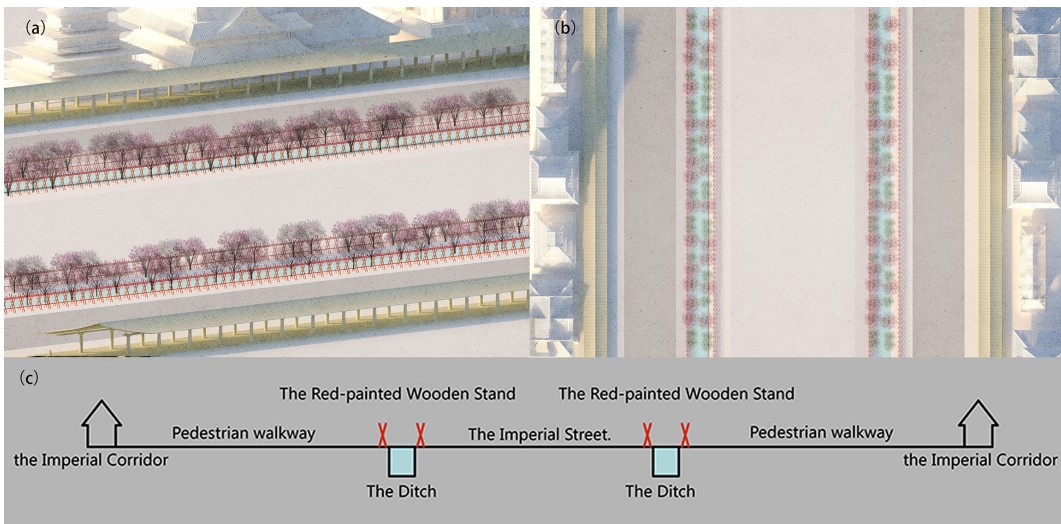

**Figure 18.** (**a**) Schematic diagram of the Imperial Street in Dongjing of the Northern Song Dynasty, (**b**) top view, (**c**) section.

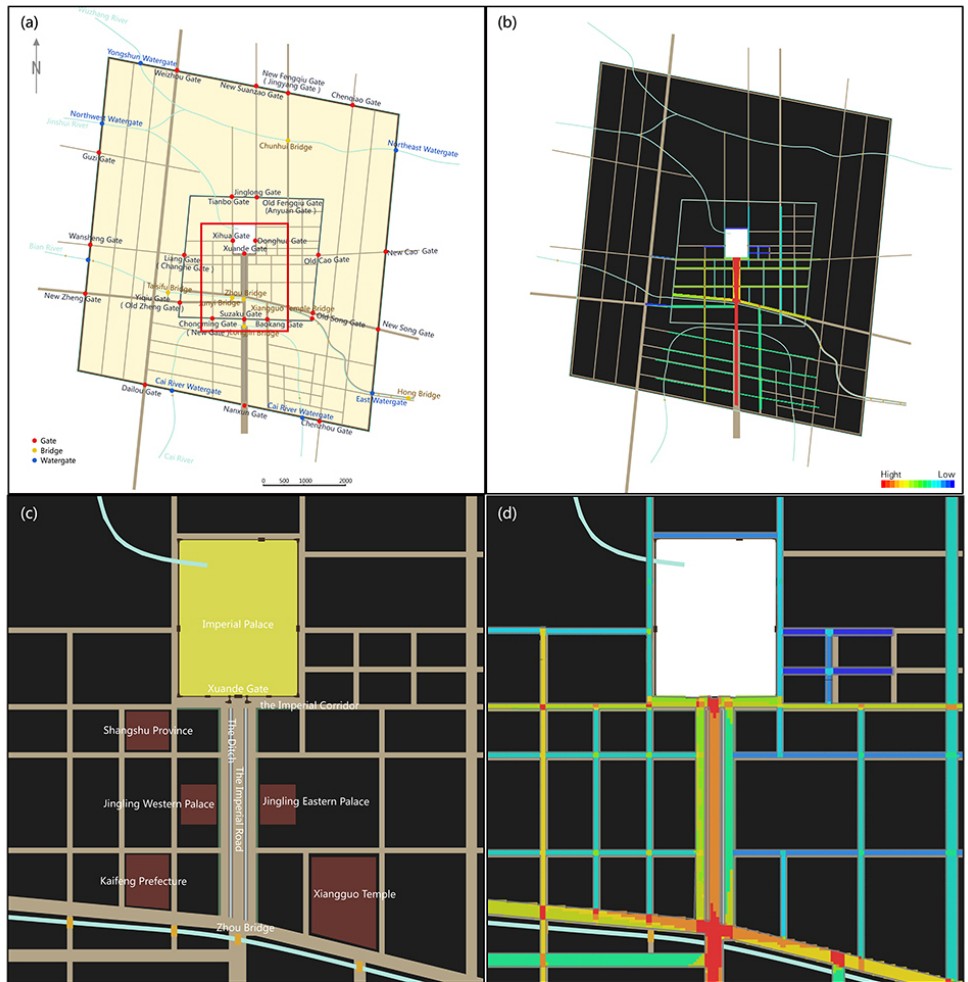

**Figure 19.** (**a**) Recovery plan of Dongjing City of the Song Dynasty (1102–1125), (**b**) the visual integration of the square in front of Xuande Gate in Dongjing City of the Song Dynasty, (**c**) the street in front of Xuande Gate,the isovist connectivity value of the square in front of Xuande Gate in Dongjing City of the Song Dynasty (**d**).

The street in front of the Xuande Gate was wider than other streets, forming a ceremonial front square. From this viewpoint, people gathered and interacted with nearby streets. Compared with other spaces, it had better accessibility, forming an open square similar to the Western square. At the end of the Northern Song Dynasty, Xuande Gate had been built in the form of five gates, with two watchtowers on both sides of the front, which enhanced its magnificent image as the Imperial Palace(Figure 20).

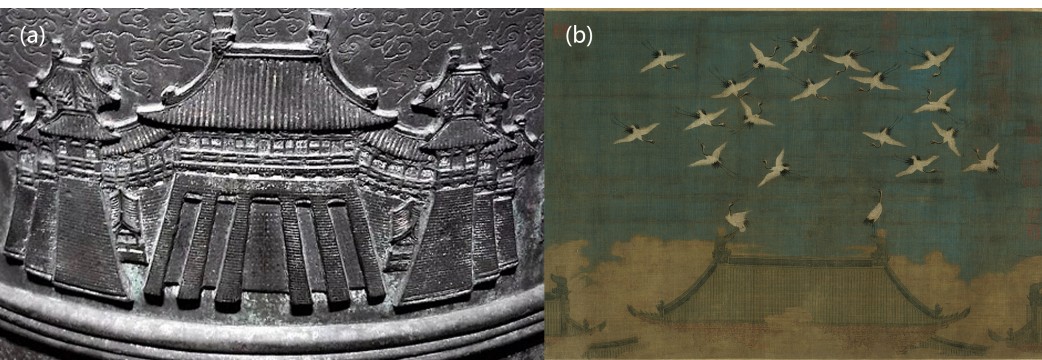

**Figure 20.** (**a**) The Xuande Gate painted on the bronze bell of Dashengfu in the Northern Song Dynasty, taken by the author at Liaoning Museum; (**b**) Auspicious Cranes.

Zhou Bridge was symbolic in Dongjing and played a different role in shaping the urban landscape. Functionally, it was the intersection of Bian River and Imperial Street; the intersection of external water transportation and the central axis's core street. It was bound to affect the surrounding businesses. There were many commercial spots, inns and commodity storage areas on both sides of Bian River. The officials and people in business from the south lived here temporarily. For a long time, this place attracted many vendors, itinerant artists, migrants and others. Finally, a prosperous commercial node with a highly dense population and commercial functions such as commerce, entertainment and culture came into being. Moreover, it formed an important spatial sequence, reflecting the traditional Chinese thoughts in the city's layout under the capital's highly centralised political system.

According to the isovist connectivity value of Figure 19d, the part in front of the Xuande Gate, and near the Zhou Bridge, was still the part with the highest connection values, where the space was more open, the view was less obstructed, and people wanted to stay here. In Imperial Street, the middle's connectivity value was higher, and the line of sight was better, and the connectivity values on both sides were lower. In this way, Imperial Street was divided into three parts that could be explained from this perspective. The Emperor had a better view of the surrounding area as he passed by, even though the middle two royal gutters were layered with lotus and pears, plums, and almonds, without completely blocking the view. Moreover, the residents are located on both sides of the walkway, making it easier for them to directly see the middle of the imperial road, which visually enhanced the royal presence's dignity.

## 5. Validation

The space syntax considers that the role between human and environment comes from spatial form; spatial form is not a static background of socio-economic activities, but a part of socio-economic activities. Therefore, the calculation results of depthmap are compared with human behaviour records to verify the validity of ancient cities' study using the space syntax. Since the research object is an ancient city, human behaviour characteristics and distribution cannot be expressed by recording people's current flow, so the calculation results are compared with those of the two capital cities of Tang and Song through resident residential distribution and commercial distribution.

The spatial distribution of businesses in Chang'an City was verified with a distribution map (Figure 21a). In the early Tang dynasty, businesses were mainly distributed in the East-

West Markets, so there was no need for verification. In the mid to late Tang dynasty, it can be seen that the distribution of commercial points had the spatial characteristics of being close to the road, adjacent to the gate and adjacent to the city. To the east of Zhuque Street, the main commercial sites were located in the Fangs of Chongren, Shengye, Pingkang, Daozheng, Changle, Xinchang, Xuanping, An-yi, Jinggong and Xingdao; to the west of the Zhuque street, and they were mainly located in the Fangs of Guanglu, Yanshou and Bujing. Most of these workshops are located on both sides of Jinguang and Chunmingmen Avenue. It was distributed around this area because this street was the main east-west street in Chang'an City, north of the Imperial City, Xingqing Palace, and along the north wall of the two cities. There were advantages in terms of location i.e., larger geographic space and connections to the two economic hinterlands of Guandong and Long-right traffic through the Chang'an City. The residential distribution map of Chang'an in Tang Dynasty basically corresponds to the location of the integrated core of Chang'an in terms of north-south distribution(Figure 21b), which were all mainly distributed around the political centre of gravity near Chunmingmen-Jinguangmen Street. However, the difference in the east-west distribution of residential houses was not reflected in the calculation results of the axis integration degree.

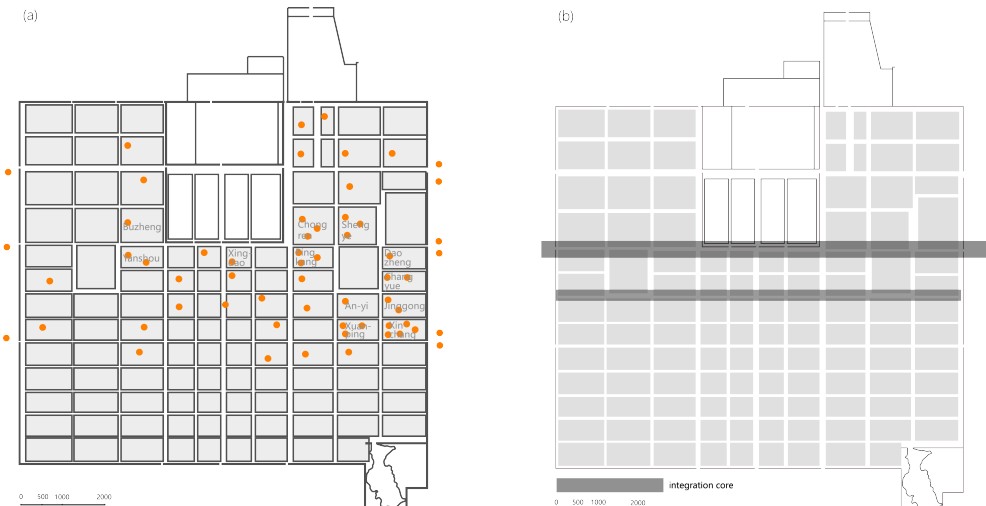

**Figure 21.** (**a**) Translated and drawn by the author from the distribution of commercial points in the Chang'an City prepared by SEO Tatsuhiko and Li Jianchao, "The Updating of the Two Capital of the Tang Dynasty [70]"; (**b**) the integration core of Chang'an.

Figure 22a shows the distribution of residences in Chang'an during the late Tang period, with 424 residences in the east of the street and only 175 in the west, which was a huge difference between east and west. Figure 22b shows the distribution of residences of officials from the An Shi Rebellion to the end of the reign of Emperor Jingzong. As the officials themselves had a certain economic base, their choice of residence was influenced by the social elements in the space, which is why their dwellings' distribution was more indicative in the argument than that of the commoners. In the figure, the residences of the officials are mainly located around the Eastern Market, which was close to the "Three Palaces", and officials lived in the vicinity of the Eastern Market for the convenience of going to court. There were several schools around the Eastern Market. The resulting diagram corresponds to the global integration of the axes of Chang'an City.

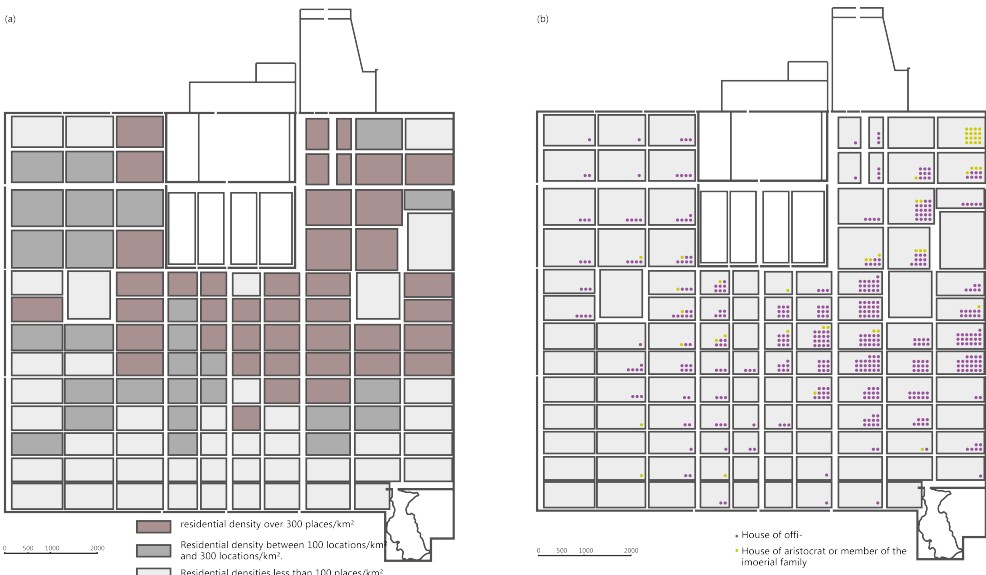

**Figure 22.** (**a**) Residential Density Distribution Map based on Zhang Yongshuai's "Study of Tang Chang'an Housing [71]". (**b**) Map of aristocratic and bureaucratic residences in Chang'an, Tang Dynasty, based on Wang Caiqiang's "Digital Reconstruction of Chang'an, Tang Dynasty [72]".

Figure 23b is the commercial distribution map of Dongjing City. In the Dongjing by Menghua Lu, the streets and lanes of nine areas are described in detail, including the Imperial Street, in front of the Xuande Gate, in and around the Zhuque Gate, near the Dongjiao Building, near the Pan Building, on the Ma Hang Street, near the Youye Gate, to the south-east of the Zhou Bridge, and near the east gate of the Xiangguosi Temple. Other documents describe Bianjing, which became Nanjing of Jin Dynasty after the fall of the Northern Song Dynasty [73,74]. Still, the available information makes it possible to establish a map of the city of Tokyo's urban layout in the late Northern Song Dynasty. In terms of the location of the commercial distribution, the commercial centres were in the southeastern part of the Imperial City, in the central business district of the Pan Building— Xiangguosi Temple, in the Ma Hang Street, in the old Caomen Street and the Baokangmen Street, all of which were densely populated with businesses. Comparing the integration core of Dongjing of the Northern Song Dynasty with the commercial distribution showed that commercial distribution was highly consistent. This indicated that the commercial distribution was jointly affected by various spatial structural factors from many aspects. (Figure 23). It further explained the space syntax model's consistency for the city's which could be interpreted as the actual situation.

The layout of the population of Dongjing city showed an unbalanced distribution. According to the number of Li-Fang and the number of inhabitants in Figure 24, the east and west sides of the city were quite unbalanced, with 33,900 households in the western part of the city and 63,850 in the east part, nearly twice as many in the eastern part. In terms of the even distribution between the east and west, the city's west side is more evenly distributed. While the east side varies considerably, with the number of households in the second left-wing and the left-wing in the east being nearly twice as many and three to four times as many as in the other quarters respectively. According to the global integration of the axes and view of the Dongjing City, the compact core is mainly located in the eastern part of the imperial city, which also corresponds to the uneven distribution of the population in space.

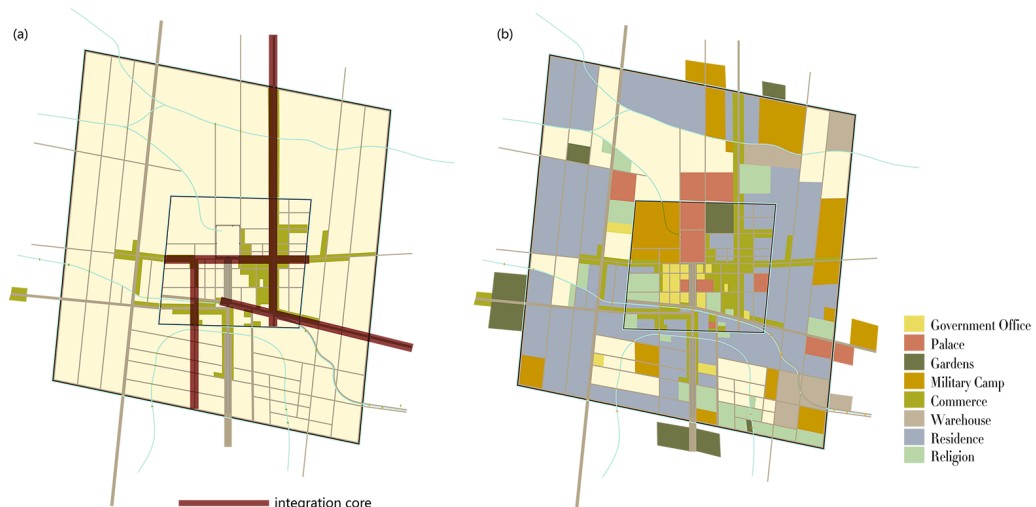

**Figure 23.** (**a**) The integration core of Dongjing city; (**b**) Functional distribution diagram of Dongjing City of the Song Dynasty based on Lu and the papers by Ye et al.

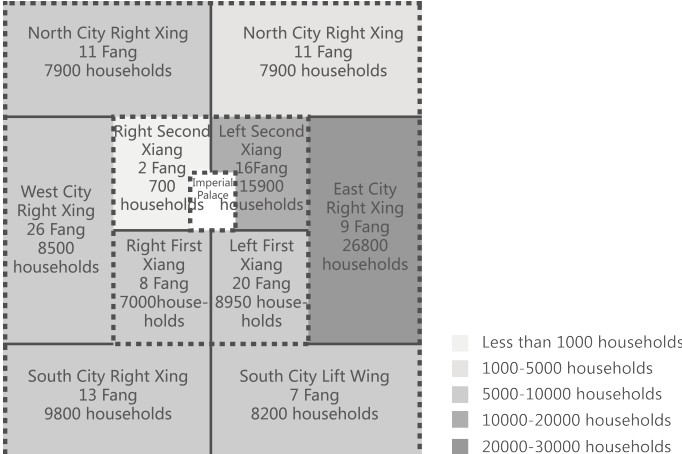

**Figure 24.** the number of Fangs and households in the city in the fifth year of the Tianxi period of the Northern Song Dynasty, with reference to the composition of the Dongjing Jie-Shi system in the Northern Song Dynasty [72].

This chapter shows the correlation by comparing the residential distribution and commercial distribution of Chang'an and Dongjing with the analysis results. It shows that the whole analysis process is correct.

## 6. Discussion

Spatial form is not a static backdrop of socio-economic activities, instead, it actively showcases spatial evolutions according to social, economic and political processes and changes. Space syntax enables the quantitate exploration of these changes and the influences of people and the environment relative to spatial form. The space syntax and depthmap analysis and findings for two ancient cities, Chang'an City and Dongjing City, were compared with documentary records to verify the validity of the space syntax findings. Since the research objects were ancient cities, human behaviour characteristics and distribution cannot be expressed by recording the current flow of people, so the resident residential distribution and commercial distribution results were compared with the historic reports of the two capital cities of Tang and Song.

The analyses show that the spatial layouts of both Chang'an and Dongjing were based on their political attributes which dominated the core of the city. Not only the formal centre, but they also dominated and controlled the spaces in the overall urban form. In comparing

the global integration value of Chang'an and Dongjing, there was an exact compact core that was located close to political and commercial spaces in both, but the morphology of the compact core varied differed considerably. Dongjing took several crossed lines as the main axes, distributed near the palace. The compact core of Chang'an had a straight axis along the east-west direction, adjacent to the Imperial Palace and the East-West Markets. This indicates that although Chang'an and Dongjing presented different forms of the compact core, they were influenced by the position of their political centres, which was inseparable from the harsh urban system implemented under the centralised imperial power of the time. The capitals are representative of the national political city, and the situation is even more outstanding. However, it can also be seen from the form of the compact core that the degree of influence of centralised rule on the urban layout was somewhat different: Chang'an City had a small number of compact cores. It showed a regular pattern, a manifestation of the strict management of urban space by the ruling class. The asymmetrical compact core of Dongjing reflected the relative free approach to urban management.

Comparing global integration and local integration, these two measures perform differently for different trip purposes. On the one hand, in the undirected path search task, local integration proves to be the most effective measure. On the other hand, in directed search tasks, the global selection is a more useful measure than local integration [75]. Comparing the local integration and the global integration in Chang'an differs significantly, which suggests a difference between purposeful and purposeless going of residents in Chang'an City. When there was a specific purpose for travel, it was usually to go to the East and West markets to make purchases or to government departments such as the Imperial City, while purposeless going usually referred to the areas of daily residents' activities, and residents mainly walked around the Fang. The global integration and local integration of Dongjing City almost overlap, showing that the residents have similar purposeful and purposeless places to go in the city, which is a result of the spontaneity of the urban layout of Dongjing City, such as business stores and entertainment industries, which have purposeful places to go distributed in places where people can gather easily and spontaneously, in places with high accessibility. Another reason is that the Song Dynasty urban policy did not strictly separate commercial and residential areas. It showed the result of consistent and mixed areas where commercial and residential were located. The above analysis shows the impact of different urban planning on residents' lives, reflected in people's travel decisions.

On intelligibility, Chang'an has a standard chessboard layout, a strict urban layout under centralised power. From the perspective of spatial topology, it is impossible to spontaneously form an apparent core in such an urban street form. According to the actual top-down management, the core of East-West Markets is not reflected in urban space's topological relationship. Thus, Chang'an was a political city. Dongjing City has a cohesive urban space with the Imperial Palace as the centre. Its political centre overlaps with its geometric centre, and its economic centre extends in all directions around the political centre. The residents' free will partly influences the streets in building and being supplemented by financial functions, so the streets were formed with a certain degree of spontaneity.

From this core, the capitals' business regions spread out from this centre. Here, the two cities also show significant differences, depending on the form of the core street. Chang'an has a regular layout, revealing strict centralised management under the constraints of the Li-Fang system, making it difficult to form an urban core from the perspective of spatial topology. Dongjing city broke away from the Li-Fang system and formed an open Jie-shi planning system, which provided a wider space for the development of the city's commercial economy and the development of social and cultural activities. This further enabled the development of commerce. The rise of public leisure and entertainment activities further contributed to the change in the city layout. Dongjing represented an existing mixed urban spatial form, encouraging the spontaneous construction and economic

behaviours of urban residents. Thus, Dongjing City demonstrates a specific degree of spontaneity and reflects the self-organising function of the city.

The comparative analysis of the commercial spaces of Chang'an City and Dongjing City, Dongjing City presents the same degree of visual integration as the global axial integration, reflecting the consistency of commercial spaces more distributed in areas with high accessibility and good visibility. The visibility graph analysis for the commercial spaces of Chang'an City did not produce corresponding results, which also shows the differences in the self-organisation of urban space between the Tang and Song capitals. The analysis also shows core spaces which not only defined the centres of the economic distribution but also assumed social and humanistic significance and became the centre of psychological belonging for the residents.

For instance, the square in front of the palace evolved from the closed etiquette space of Chang'an into the public entertainment space of Dongjing City. Judging from the main characters in the square, the main political participants of Chang'an were the emperor and his ministers. However, its function as a place of entertainment for citizens can only be reflected in festivals. In Dongjing, the people frequently engage in activities in the squares' streets and alleys. Such public open space affects the city from the spatial order and provides citizens with space for mutual communication and public entertainment, making the city more relaxed and humanistic.

The visibility analysis demonstrated that the relationship between the city and people will be infinitely weakened when people's movement does not influence the city's growth and change. However, it is also possible that people's life path does not always conform to the urban network. Therefore, these activities likely took place in the streets within the residences that were not explored in this study. In Chang'an for instance, ordinary residents spend their daily lives within the lanes. Therefore, the direction of roads and the arrangement of houses will inevitably better represent the residents' living conditions. In Dongjing, nodes were found that attracted all kinds of commercial activities. The streets and alleys with low viewshed integration value were mostly found in the internal second-level streets and alleys. These areas have fewer intersection points. In such areas, it is difficult to observe other spaces, resulting in poor visibility. However, these findings represent a limitation in the granularity of viewshed analysis, not as the tool's failure but to signpost the researchers' attention to further explore the underlying causes.

### 7. Conclusions

The space syntax method enabled a quantitative representation of the spatial composition and usage of the explored ancient cities. It facilitated the description and analysis of the relationship between physical space and social elements and provided tools to measure the city's network of street spaces. Although the space syntax has been developed based on the study of modern urban planning, the use of a common methodology in the study of ancient and modern cities may lead to a common language in the study of cities. This, in turn, will result in new insights into the urban space, and a holistic study of ancient urban spaces of the past, that cannot be achieved by archaeology and literature review alone.

The method was used to study and compare based on the close relationship between urban network structure and human movement in Chang'an and Dongjing, making it feasible to transform findings into verifiable theories. It also made it possible to identify spatial configurations that cannot be seen in street plans based on qualitative analysis alone.

Some limitations in the analysis of space syntax were found, such as the inability to connect the calculation results with reality and the bias of the results due to the lack of urban information. Nevertheless, as a new quantitative spatial analysis method, space syntax provides a new perspective for analysing the spatial structure of ancient cities.

**Author Contributions:** Conceptualization, L.Y. and T.W.; methodology, L.Y.; software, L.Y.; validation, L.Y.; formal analysis, L.Y.; investigation, L.Y.; resources, L.Y.; data curation, L.Y.; writing—original draft preparation, L.Y.; writing—review and editing, L.Y., T.W. and K.A.; visualization, L.Y.; supervision, T.W.; project administration, T.W.; funding acquisition, T.W. All authors have read and agreed to the published version of the manuscript.

**Funding:** National Social Science Foundation of China (Grant No.17BH175).

**Institutional Review Board Statement:** Not applicable.

**Informed Consent Statement:** Not applicable.

**Data Availability Statement:** Data available in a publicly accessible repository.

**Conflicts of Interest:** No conflict of interest.

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
