# Peer review of "A Comparative Study of Urban Spatial Characteristics of the Capitals of Tang and Song Dynasties Based on Space Syntax"

_urbansci, doi:10.3390/urbansci5020034_

Round 1
Reviewer 1 Report
-This manuscript employs the space syntax theory for a comparative study of the urban spatial structure in two capitals in China’s ancient feudal society. The manuscript is well written and supported by good displays. Yet, I have some remakes that need clarifications before recommending it for publication.
-The authors should clearly list their research questions and show the novelty of their work in the introduction section.
-The authors use syntax theory to explore the urban spatial structure. I would recommend authors to briefly discuss and compare syntax theory with other approaches that used to study the urban spatial structure and/or functions in general such as spatial matrices and regression models (recommended examples: Liu et al., 2017; Mustafa et al., 2018; Mustafa and Teller, 2020).
>Liu, H., Huang, X., Wen, D., Li, J., 2017. The Use of Landscape Metrics and Transfer Learning to Explore Urban Villages in China. Remote Sensing 9, 365.
>Mustafa, A., Saadi, I., Cools, M., Teller, J., 2018. Understanding urban development types and drivers in Wallonia. A multi-density approach. International Journal of Business Intelligence and Data Mining 13, 309–330.
>Mustafa, A., Teller, J., 2020. Self-Reinforcing Processes Governing Urban Sprawl in Belgium: Evidence over Six Decades. Sustainability 12, 4097.
-Line 43: “The quantitative analysis makes the analysis process more convenient, and the conclusions drawn are clearer and more intuitive.” The authors should support this claim by further explanation/discussion.
-Line 82: “Space syntax focuses on the analysis of the structural system of spaces composed of the connection relationship between its nodes. The quantitative analysis is carried out by the space syntax software Depthmap. There are three basic application methods of space syntax: convex space, axial line and visibility graph analysis.
The more common basic syntax variables include connectivity value, integration value, intelligibility and synergy.
The connectivity value is the number of spaces that intersects with other spaces. The higher the connectivity value, the better the spatial permeability. The integration value represents the relationship between a space and its local space or global space. The higher the integration value, the higher the accessibility. Intelligibility is the linear correlation value between the connectivity value and the global integration value. Synergy is the linear correlation value between the global integration value and the local integration value[20].” This part doesn’t read clear. Please rephrase.
-The authors did not provide the results section. They instead presented their analyses within the methods section. Why?.
-How did you validate/evaluate your analyses?
-Line 239: “The intelligibility index of Chang’an is 0.170 (Figure 9a). The city with low intelligibility indicates that the local space is not closely integrated with the structure of the whole. The accuracy of predicting the whole through local space is low, and the local space in the Fang is disconnected from the overall urban system. The reason being that each Fang is rectangular with two or four gates through which the residents of the Fang must enter.” The intelligibility index of 0.17 is too low. Can you provide more explanation, evidence, and references?
-English needs improvements.
Reviewer 2 Report
I commend the authors for the choice of case studies, and for re-creating the plans for both sites. Please find below some comments:
- Minor comments:
- The abstract needs to be re-written to better introduce the methods, namely space syntax, and make the conclusions concise.
- In the abstract: the authors suggest an evidence between structure and human behavior. However, I did not see that in the paper.
- Hillier and Hanson should be credited for Space Syntax (and not Hillier alone).
- Space syntax was not properly introduced and the literature review lacks reviews of findings from previous studies.
- There was no proper introduction to Depthmap (credits must be given to Turner et al.)
- The citations are not consistent
- Major comments:
- The paper lacks a good structure. Maybe stating with a description of the site, followed by description of space syntax methods and measures, description of how the authors will analyze and compare sites.
- There was no introduction to visibility graph analysis. Michael Benedikt, the originator of isovist concept, must be credited (the concept of isovists was not developed by the space syntax group). I recommend readings by Jean Wineman, Michael Benedikt, Ruth Dalton, Linda Nubani, Mahbub Rashid
- My most concern revolves around the author's computation of VGA. VGA is calculated based on visible spaces as bounded by physical environment (such as buildings, trees, fences). I examined the VGA in this paper and it appeared to me that it was only calculated based on street spaces but without considering other physical barriers (this was not acknowledged). Therefore, I am afraid that none of the results from this section are valid (unless the authors did compute barriers)
Round 2
Reviewer 1 Report
I would suggest moving the validation of the analysis to the result section. That makes more sense. Also, the authors still need to clarify the validation procedures.
Author Response
Re: thank you for the offer.
Have put the validation part of the analysis in the results section.
I have described the validation process briefly in the first paragraph of the validation (lines 449-455) and depicted the validation process in detail in lines 456-507. It is also briefly described in the introduction section (lines 56-62).
Reviewer 2 Report
- The abstract still needs editing and structure (preferably use professional editorial services). The author should report what the results/ findings were. For example, was the author trying to compare the distribution of residences and commercial points across two different time periods using space syntax? If so, please amend and update. On what basis was the conclusion made around the political impact on layout? This needs to be clarified in abstract and content.
- Your revised credits now says "Hillier, Hanson et al", it should be only Hillier and Hanson (no others)
- I still feel that space syntax is not properly introduced. The definition restricts this method to urban streets only (space syntax can
be applied to any environment, indoor and outdoor). I have provided to the readers some good references to help, I don't see those in the list. - 100 to 110 contains false information. I urge the authors to read papers by Alasdair Turner and give him the appropriate credit
(for example, he was not credited to be the founder of space syntax). Also, Depthmap is just a software, one of many, that is used to
apply space syntax. The definition is very confusing - Line 110, a field of view was not developed by Benedikt. He advanced the use of isovists, etc. The use of "field of view" throughout this paragraph is odd and inappropriate
- What is feasible integration?
- The labels of figure 15 is confusing. It will be very helpful if the authors put a time stamp or uses a consistent description of
whether the plan is old or new (which periods they belong to) - The response by the author is confusing, if the information is not available (which is understandable) then doing a VGA is not possible. The author's response acknowledges including trees, buildings and railings but i don't see this in any of the figures. I still have problems accepted a VGA run on streets only (methodologically, it is incorrect)
- This should be a good paper. I highly recommend the authors to take their time in re-drafting the manuscript.
Round 3
Reviewer 2 Report
This statement "According to the representation, a building plane is divided into a finer grid, e.g. 100 x 100. Each fin r cell representsa single vantage point and an associated isovist" is confusing and makes it sound like the definition of isovists while it is really an explanation of how depthmap works
There is no such a thing called "Integration degrees are divided into global and local integration degrees". You may want to use values instead.
Revise this language "Intelligibility is a linear correlation value indicating the degree of connectivity and global integration" and the rest of the paragraph (repetitive) as it is mentioned twice
When you say "Using the spatial axis and visual field diagram in Depthmap" are you referring to Visibility Graph Analysis? Because VGA and Visual field are two different things
My early feedback is still not met and I don't understand what your response was. So please let me explain couple of things:
- All figures look like they were axial line analysis (I didn't see any Visibility Graph Analysis) which I assume you are using Visual Field instead. It is not too clear if "integration" was the result of axial line analysis or visibility graph analysis.
- If you were trying to tell me that you did perform a visibility graph analysis, it really doesn't look like one. To me, all the black shading in the figure represents the outline of the building, is this correct? Are all the buildings located perfectly on the streets with no gaps? This is what I need the answer to? This is a methodological concern. The same applies for the axial line analysis because the axial lines should be long, intersect with one another, broken into smaller long lines of sight
- Figure 13 caption reads "Plan view of Chang’an (a), visual integration (b), visual integration near the greenery (c)" What does C represent, is this a section view?! What do you mean by greenery? Why is the Imperial City greyed out? I did see the close up of the East Market, again, is the East Market surrounded by high walls or gates all around?
